# A single touch can provide sufficient mechanical stimulation to trigger Venus flytrap closure

Jan T. Burri[1☉], Eashan Saikia[2☉], Nino F. Läubli[1☉], Hannes Vogler[3☉], Falk K. Wittel[2], Markus Rüggeberg[2,4], Hans J. Herrmann[2¤], Ingo Burgert[2,4], Bradley J. Nelson[1]*, Ueli Grossniklaus[3]*

1 Department of Mechanical and Process Engineering, ETH Zurich, Zurich, Switzerland, 2 Department of Civil, Environmental and Geomatic Engineering, ETH Zurich, Zurich, Switzerland, 3 Department of Plant and Microbial Biology and Zurich-Basel Plant Science Center, University of Zurich, Zurich, Switzerland, 4 Swiss Federal Laboratories for Materials Science and Technology—EMPA, Cellulose & Wood Materials Laboratory, Dübendorf, Switzerland

☉ These authors contributed equally to this work.
¤ Current address: Laboratoire de Physique et Mécanique des Milieux Hétérogènes, École Supérieure de Physique et de Chimie Industrielles de la Ville de Paris, Paris, France
* grossnik@botinst.uzh.ch (UG); bnelson@ethz.ch (BJN)

**Data Availability Statement:** All relevant data are within the paper and its Supporting Information files. Data underlying the plots are available in the supporting data file S1 Data. The source code for

## Abstract

The carnivorous Venus flytrap catches prey by an ingenious snapping mechanism. Based on work over nearly 200 years, it has become generally accepted that two touches of the trap's sensory hairs within 30 s, each one generating an action potential, are required to trigger closure of the trap. We developed an electromechanical model, which, however, suggests that under certain circumstances one touch is sufficient to generate two action potentials. Using a force-sensing microrobotic system, we precisely quantified the sensory-hair deflection parameters necessary to trigger trap closure and correlated them with the elicited action potentials in vivo. Our results confirm the model's predictions, suggesting that the Venus flytrap may be adapted to a wider range of prey movements than previously assumed.

## Introduction

The hunting mechanism of the carnivorous Venus flytrap (*Dionaea muscipula*), according to Darwin "the most wonderful plant in the world" [1], has attracted the interest of many scientists, starting with the observations made by Edwards and Nutall, who described the excitability of the sensory hairs but still thought that the capture of insects was accidental [2,3]. Only in the 1830s, Curtis realized that the traps were specifically devoted to catching animal prey [4]. Since then, the individual phases—from trap triggering to reopening after successful digestion —have been investigated from different angles (reviewed in [5]). Starving plants attract insects through the secretion of volatile compounds [6]. While exploring the trap for food, wandering insects accidentally touch one of the six sensory hairs distributed on the two lobes of the trap,

the ECB model is available on GitHub (https://github.com/wittelf/ECB) as well as via Zenodo (https://zenodo.org/record/3799874#.XrIEBi1XbFw) and can be cited by using the DOI 10.5281/zenodo.3799873 (all versions).

**Funding:** This work was supported by the University of Zurich, the ETH Zurich, and a grant from the Swiss National Science Foundation (Interdisciplinary Grant CR22I2_166110) to UG, BJN, and HJH. The funders had no role in study design, data collection and analysis, decision to publish, or preparation of the manuscript.

**Competing interests:** The authors have declared that no competing interests exist.

**Abbreviations:** AP, action potential; ECB, electromechanical charge buildup; MEMS, microelectromechanical system; RP, receptor potential.

thereby triggering an action potential (AP) [7–12]. A second touch-triggered AP within about 30 s causes the trap to snap, and the prey is caught. Further APs triggered by the struggling prey induce jasmonic acid biosynthesis and signaling [13, 14], which seals the trap tightly and eventually leads to the formation of the "green stomach," a digestive cocktail that mobilizes prey-derived nutrients [12, 15, 16].

Here, we focus on the translation of the mechanical stimulation of the sensory hairs into an electrical signal. Although there is a general agreement that sensory-hair deflection opens mechanosensitive ion channels, such channels have not yet been identified [17, 18]. While these putative channels are open, a receptor potential (RP) builds up [10, 19], and, if the deflection is large enough, the RP reaches a threshold above which an AP is elicited. Previous attempts to correlate the mechanical stimuli to the generation of APs suffered from the lack of appropriate instrumentation [10, 19] and thus were not quantitative, and/or the experiments were done in fixated or dissected, nonfunctional traps [20]. To overcome these shortcomings, we used a microelectromechanical system (MEMS)-based force sensor mounted on a microrobotic system to precisely control the velocity and amplitude of the deflection and to simultaneously measure the applied force in vivo (Figs 1A and 1B and 2). In this way, we were able to accurately quantify the parameter range in which hair deflection leads to trap closure, while a second force sensor measured the generated snap force (Fig 2). In addition, using a noninvasive method, we measured APs to test the deflection conditions under which they are generated.

## Results

### Two fast, consecutive deflections trigger trap snapping if a certain angular displacement or torque is reached

Reasoning that hair deflection induced by spiders, ants, and flies—i.e., the "classical" prey of *D. muscipula* [21]—would be rather quick, we operated the microrobotic system at full speed to simulate these stimuli, resulting in high initial angular velocities ranging from 10 to 20 rad s$^{-1}$. This is in the same range as Scherzer and colleagues found for moving ants, which deflect the sensory hair with an angular velocity of 0.25–7.8 rad s$^{-1}$ [20] but much slower than the leg movements of houseflies [22]. At such high velocities, the duration of a deflection is much shorter than other involved time-dependent factors, such as the decay of the RP [19] and the relaxation of the sensory hair (Fig 1F). Considering angular rather than linear deflection allowed us to correct for differences in the contact height of the sensor probe relative to the constriction site of the sensory hair as well as for different sensory-hair geometries (Figs 1B, 2C and 2D). Therefore, a single deflection can be approximated by a discrete increase in the angular displacement, and the triggering of an AP mostly depends on the magnitude of the angular displacement. We defined a single deflection as the combination of a back-and-forth angular displacement, similar to what happens when an animal touches the hair. Each measurement consisted of two subsequent deflections with a 1-s gap between them, up to a predefined angular displacement θ. If the trap did not close, we waited for 2 min to make sure that the RP was completely reset. Since RP measurements are destructive and not compatible with our noninvasive in vivo experiments, we relied on data from the literature [10, 19]. The duration of this waiting period was chosen because a series of earlier experiments had demonstrated that, at temperatures below 30°C, two deflections within 30–40 s were necessary to rapidly and completely close the trap. Although a summation effect was reported in these publications, at least five deflections spaced by 2 min were necessary to induce trap movements, which were always slow and/or partial [23–25]. After the dwell time, the procedure was repeated with increasing angular displacements until the snapping mechanism was triggered

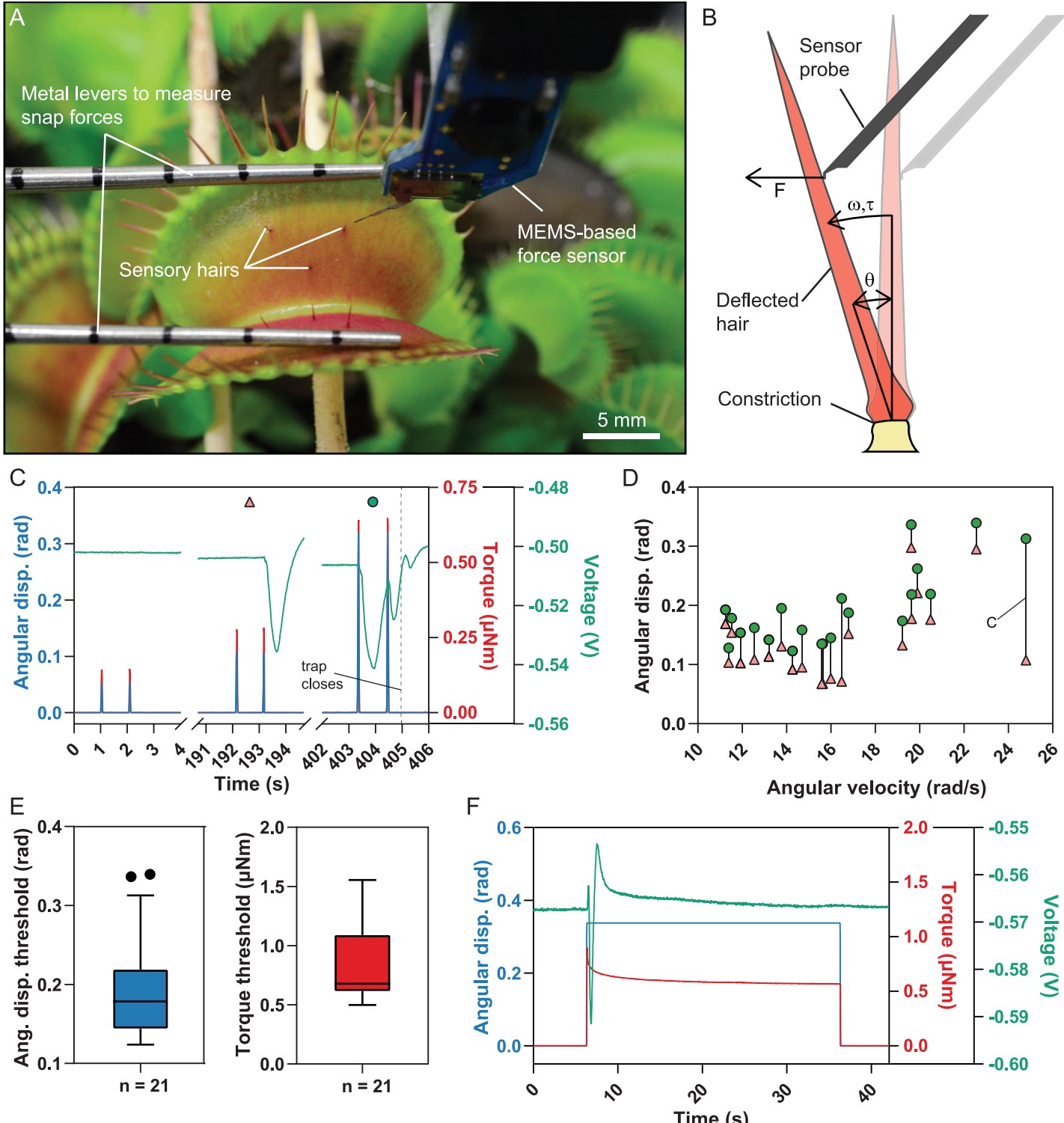

**Fig 1. Double deflection and sustained displacement of sensory hairs.** (A) Experimental setup showing a MEMS-based force sensor placed next to a sensory hair. The metal levers prevent the trap to close upon triggering and simultaneously measure the snap force via a load cell. (B) Scheme of a deflection experiment. The sensory hair is deflected by a linear movement of the sensor probe whereby the force $F$ is measured. Angular displacement $\theta$, angular velocity $\omega$, and the torque $\tau$ can be determined (for details, see Materials and methods and Fig 2D). (C) Successive double deflections of the sensory hair at increasing angular displacement ("disp.") with recorded torque and voltage. (D) Individual measurements showing the double deflection that led to trap triggering (green circles) and the preceding, nontriggering one (red triangles). The experiment shown in (C) is indicated. (E) Descriptive statistics of the angular displacement and torque threshold for double deflections that initiated trap

closure. The horizontal line indicates the median, the box the 25th and 75th percentiles, and the whiskers extend to the most extreme data points without outliers. (F) Angular displacement of the sensory hair sustained for 30 s with recorded torque and voltage. For the underlying data see sheets Fig 1C to Fig 1F in S1 Data. MEMS, microelectromechanical system.

(Fig 1C and 1D), which was the case when a median displacement threshold of $\theta$ = 0.18 rad or a median torque threshold of $\tau$ = 0.8 μN m ($n$ = 21) was reached. We never observed trap closure below $\theta$ = 0.12 rad and $\tau$ = 0.50μN m (Fig 1E and S1A Fig), such that this represents the lower limit of angular deflection, which is necessary to trigger trap closure under our conditions. Using the torque threshold, we determined that an animal has to apply twice a force $F$ around 0.5 mN close to the tip of a sensory hair with a length of 2 mm, or up to 5 mN close to the constriction (S1B Fig), to trigger closure.

AP measurements provide the link between sensory-hair deflection and trap closure. When the two consecutive deflections were well below the displacement threshold ($\theta \ll 0.12$ rad), we never observed an AP. For deflection amplitudes near the displacement threshold ($\theta < 0.12$ rad), a single AP was elicited after the second deflection. This indicated that both deflections contribute to the RP and that the AP induction threshold was only reached with the second

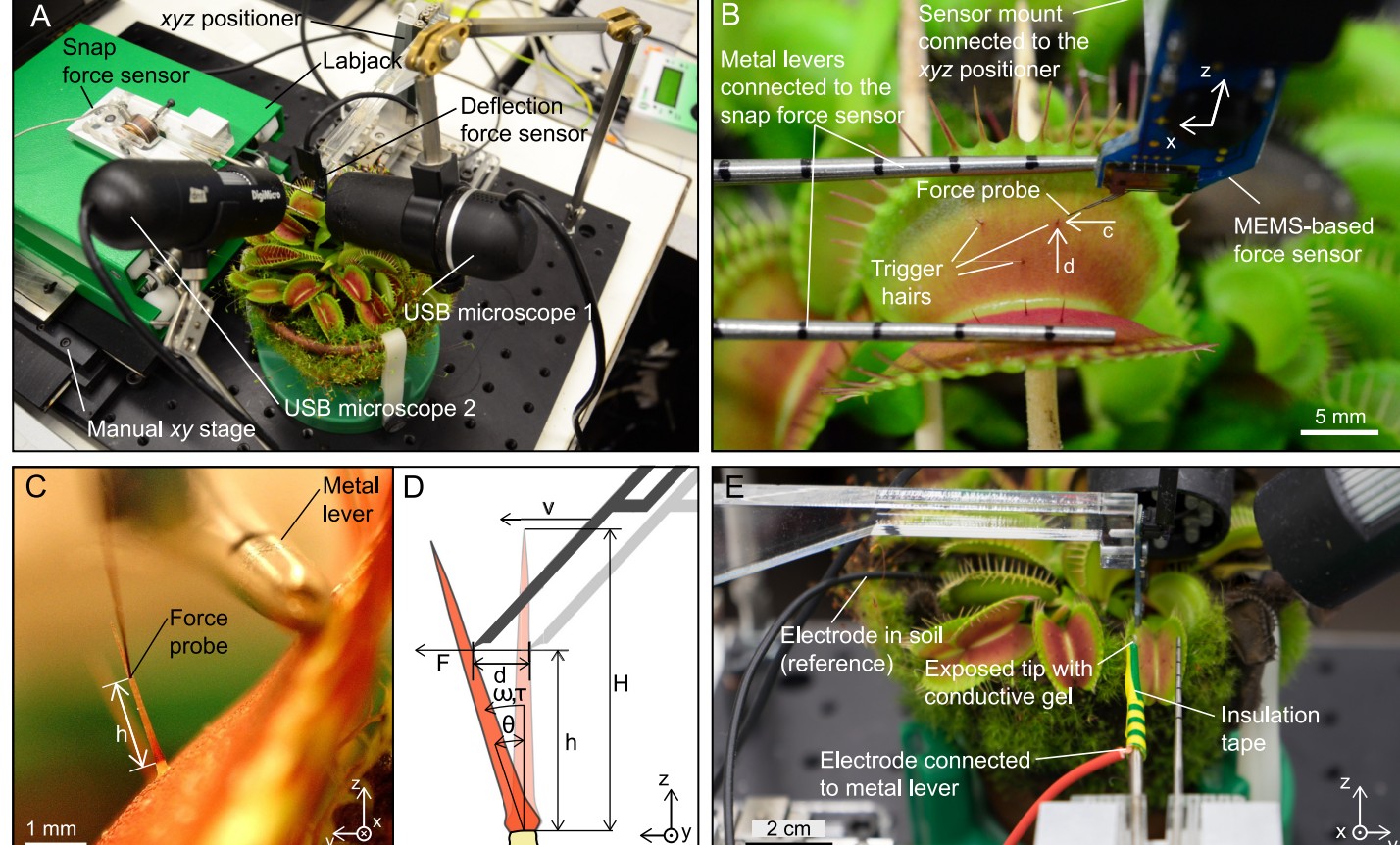

**Fig 2. System configuration to investigate the necessary mechanical stimulation of the sensory hair triggering trap closure while measuring the resulting forces and APs.** (A) The system combines a force probe to precisely control the deflection of sensory hairs and a second force sensor that measures the subsequent snap forces. Two USB microscopes are used to observe the experiment. (B) Close-up of the deflection force sensor used to deflect the sensory hair in direction parallel to the midrib. (C) View from the USB microscope 1 (on the right in A) used to extract the geometry of the sensory hair. (D) Schematic side view of hair deflection where the velocity $v$, the deflection $d$, and the force $F$ are controlled and monitored. (E) For AP measurements, one electrode was connected to the metal lever and the other one inserted into the soil. AP, action potential; MEMS, microelectromechanical system; USB, universal serial bus.

deflection. As expected, a single AP was not sufficient to trigger trap closure. The assumption that each touch triggers an AP only applied if the sensory-hair deflection exceeded the displacement threshold. If this was the case, two APs, one for each deflection, were generated and led to trap closure (Fig 1C and S1 Video).

These results suggest that a fast deflection of the sensory hair increases the RP to a certain level, which depends on the amplitude of the angular deflection. RPs can add up and may elicit an AP after several deflections if they are below the deflection threshold. However, the generation of one AP per touch only holds true if sensory-hair deflection is above the deflection threshold.

## Sustained angular displacement does not trigger trap closure

Since the increase of the RP from multiple deflections is additive, the question arose whether a sustained displacement had a similar effect. To test this, we deflected the sensory hairs far beyond the angular displacement threshold and kept that position for 30 s (Fig 1F). None of the traps closed during sustained displacement ($n = 11$, $\theta = 0.31$ rad $\pm 0.07$ rad, mean $\pm$ standard deviation). The initial displacement elicited a single AP, after which the voltage quickly returned to the baseline, despite the hair staying deflected. If sustained displacement had contributed to the RP, it should have remained above the threshold, in which case we would expect a series of APs. Our observations, in agreement with Jacobson [10], confirm that a change in angular displacement (i.e., the deflection rate) plays a pivotal role to build up the RP, whereas static deflections do not contribute. This is in contrast to an earlier finding in which prolonged manual displacement led to trap closure [26]. However, the oscillations that come with manual hair deflection are probably larger than the angular displacement threshold and may therefore represent multiple displacements rather than a single sustained displacement.

## An electromechanical charge buildup (ECB) model predicts single-touch trap snapping

Based on these findings, we developed a simple model to explore the limits of angular displacement and velocity within which the traps would react. In our ECB model, a mechanical deflection leads to a charge buildup of the RP as a function of the angular velocity $\omega$ and the displacement $\theta$, while the charges continuously dissipate. If the accumulated charges surpass a certain threshold value $Q_{RP}^{th}$, an AP is elicited. Additionally, we implemented a refractory period $t_{RP}$, representing the time interval required before another AP can be induced [27, 28]. In the Venus flytrap, subsequent APs separated as close as 0.75–1 s were reported [16, 26]. The model reproduced the experimentally found bounds for double deflections and predicted that if the deflections are too fast and/or too small, more than two deflections might be required to trigger trap closure (red area in Fig 3A) because a single deflection is not sufficient to elicit an AP (Fig 3B), something we also observed in our experiments (Fig 1C, middle). Similarly, the model showed that very low angular velocities ($\omega < 0.04$ rad s$^{-1}$) cannot fill up the RP. Unexpectedly, the model predicted a range of intermediate angular velocities (0.04 rad s$^{-1}$ $< \omega <$ 10 rad s$^{-1}$), at which a single deflection is sufficient to elicit the two or more APs that are necessary to initiate trap closure (Fig 3B).

## A single deflection at intermediate angular velocity triggers trap closure

As single-deflection trap triggering is contrary to the accepted view, we wanted to experimentally test this prediction of the model. Indeed, we observed trap closure induced by single deflections at lower angular velocities (S2 Video). To narrow down the range in which this occurs, we repeatedly deflected the same sensory hair with varying angular velocities until the trap closed. Between two consecutive deflections, we waited 2 min for the trap to recover and any RP to dissipate.

The lower boundary of the angular velocity $\omega$ required for single-deflection closure was determined by incrementally increasing it after every deflection ($n = 17$). An initial velocity below 0.009 rad s$^{-1}$ was chosen, which never resulted in trap closure. Subsequent single deflections were performed at increasing velocities until the trap shut (Fig 4A). The higher boundary was similarly determined by starting with a velocity $\omega > 3$ rad s$^{-1}$, followed by a stepwise decrease until the trap shut ($n = 9$). Additionally, we performed another set of single-deflection experiments ($n = 5$), in which the speed of the force probe was kept constant, leading to an intermediate angular velocity between 0.2 and 0.4 rad s$^{-1}$, while the angular displacement $\theta$ was gradually increased during subsequent deflections in order to get the lower boundary of $\theta$ required to trigger trap closure by a single deflection.

All the single deflections resulting in trap closure, together with the preceding stimuli for which no trap closure was observed, define the region where a single deflection triggers closure (Fig 4B and S1C Fig). The model output for single-deflection triggers covers a similar parameter space as experimentally obtained for angular deflection $\theta$ versus angular velocity $\omega$. We observed that a single deflection can trigger trap closure at intermediate angular velocities of the deflection (0.03 rad s$^{-1} \leq \omega \leq 4$ rad s$^{-1}$) but is not sufficient at slower or faster angular velocities (see S2 Fig for experimental data on single and double deflections). Incorporating the single-deflection experiments into the model provides a better prediction of the area in which a single touch can lead to two APs and thus trap closure (Fig 3A).

In the slow-velocity deflection experiments, we were able to determine whether trap triggering happened while advancing ($n = 6$) or retracting ($n = 11$) the force probe. Our results indicate that both back and forward displacements contribute to the overall RP level. AP measurements disclosed how these single deflections lead to trap closure. When the trap closed during the advancement of the force probe, two APs, one shortly after the other, were observed during the bending of the hair (advancing the probe) (Fig 4C and S3 Video). In the second case (triggering during retraction), one AP was fired during initial bending and a second while retracting the sensor probe as the hair returned to its original position (Fig 4D). In both cases, the second AP led to immediate trap closure.

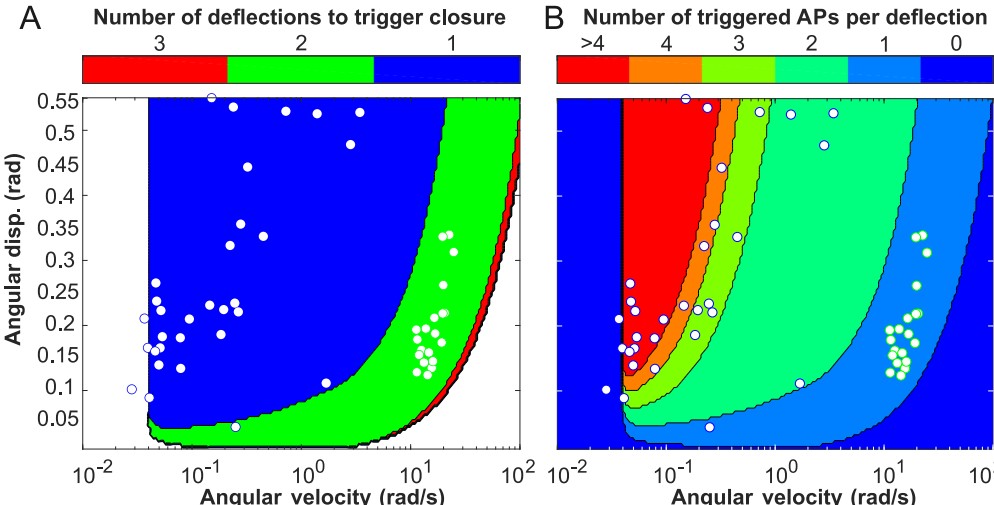

**Fig 3. ECB model.** (A) Prediction of the number of action potentials triggered for a single deflection with consecutive waiting time as function of angular displacement ("disp.") $\theta$ and velocity $\omega$, confronted with experimental observations for cases in which trap closure was triggered by single (blue circles) and double deflections (green circles). (B) Number of deflections necessary to trigger trap closure with phases of one, two, or three deflections compared to the findings for single- (blue circles) and double-deflection (green circles) experiments. For the underlying data, see sheet Figs 3A and B in S1 Data. ECB, electromechanical charge buildup.

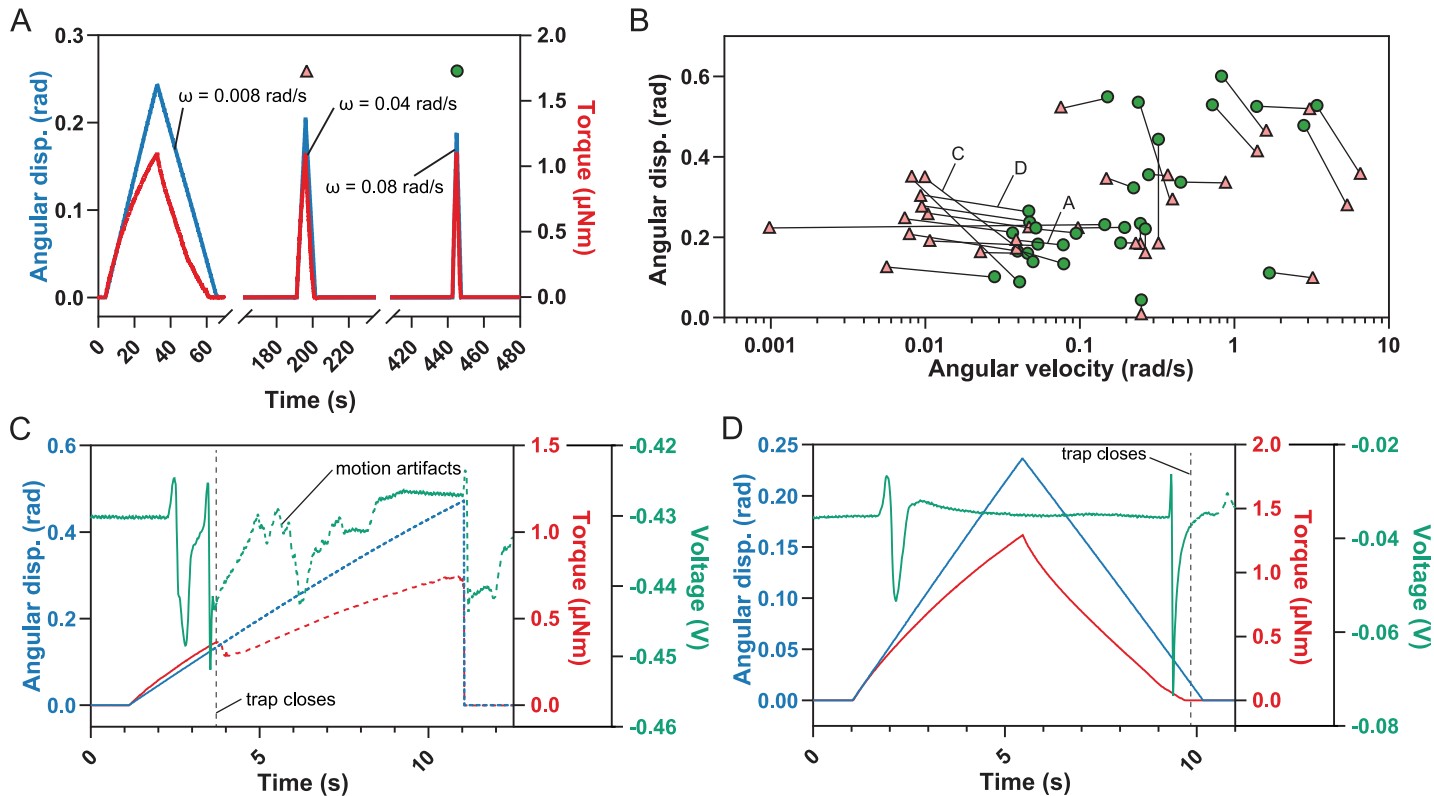

**Fig 4. Slow single deflections lead to trap closure.** (A) Successive single displacements ("disp.") of the sensory hair with increasing angular velocity while the torque is recorded. (B) Individual single-deflection experiments showing the stimulus that initiated trap closure (green circles) and the preceding, nontriggering deflection (red triangles). The specific experiments shown in (A), (C), and (D) are indicated. (C and D) Single deflections at an intermediate velocity that led to trap triggering where either (C) two APs were generated during bending (on the way in), or (D) one AP is fired at the beginning of bending (on the way in) and a second at the end of the return (on the way out). Only the data up to the start of the trap closure were used. The data affected by motion artifacts are indicated by dotted lines. For the underlying data, see sheets Fig 4A to Fig 4D in S1 Data. AP, action potential.

## Quantification of the trap snap forces and torques

In addition to the deflection parameters, we also measured the snap force developed by the triggered trap (Fig 5A). A median snap force $F_{close}$ of 73 mN was determined from 48 different traps (Fig 5B), ranging from 18 to 174 mN in extreme cases. These values are slightly lower than the previously published 140–150 mN measured with a piezoelectric sensor film [29, 30]; however, we measured the force at the beginning of snapping while Volkov and colleagues measured the impact of the lobe rims. Because the measured force strongly depends on the force sensor's position, as well as on the orientation and size of the leaf, the closing torque around the midrib $\tau_{close}$ with a median of 0.65 mN m is a better quantity to characterize the trap's closing force (Fig 5C). The delay time, i.e., the time between the mechanical stimulus and the start of trap closure, was 0.6 ± 0.3 s (mean ± standard deviation, $n$ = 18) based on video analysis, which is in the same range as the 0.4 s previously reported [31].

## Discussion

We precisely quantified the mechanical parameters that play a role in the Venus flytrap's snapping mechanism in vivo and correlated them with the electrical signals that transduce this touch information (Fig 6).

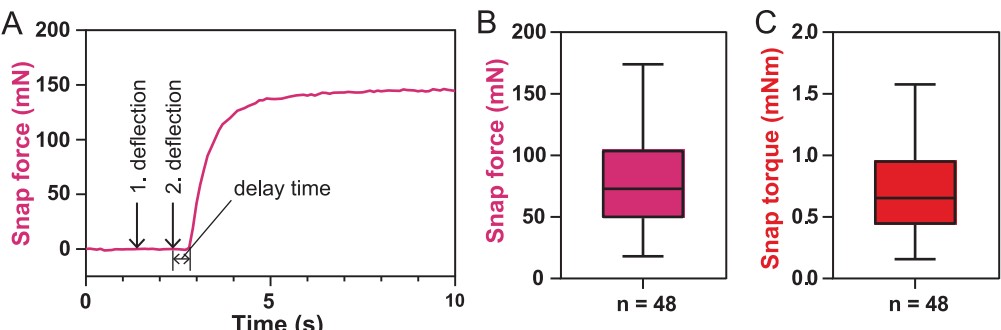

**Fig 5. Snap force and torque measured with the snap force sensor.** (A) Typical force curve after trap closure induced by double deflection. (B) Descriptive statistics of the snap force right below the marginal teeth and (C) the closing torque around the midrib of a triggered flytrap. The horizontal line indicates the median, the box the 25th and 75th percentiles, and the whiskers extend to the most extreme data points. For the underlying data, see sheets Fig 5A to Fig 5C in S1 Data.

**Fig 6. Schematic summary of the closing behavior of a Venus flytrap when its sensory hair is subjected to various controlled mechanical stimuli.** (A) A low angular velocity cannot fill up the RP and does not create an AP. (B) A deflection at an intermediate angular velocity keeps the RP close to its threshold and can lead to the release of two APs triggering the trap. (C) A fast deflection results in an AP if the deflection is larger than the angular displacement threshold $\theta_{th}$. If this deflection is repeated a second time within about 30 s, a second AP will result in trap closure. Repeating deflection below $\theta_{th}$ can fill up the RP to emit an AP. (D) Sustained displacement is a combination of a fast displacement as in (C) with keeping the sensory hair bent at a constant displacement. A single AP is generated during the initial step. While the sensory hair is kept deformed, however, no RP is built up, and the trap will not close. AP, action potential; RP, receptor potential.

Our mechanical data are largely in agreement with those recently published by Scherzer and colleagues, although in their experiments, the sensory hairs seem to be a bit more sensitive than in ours. This could be due to several reasons, such as differences in sensor calibration methods or the environmental conditions during the experiments. It has been documented that the trap closure mechanism is more sensitive at higher temperatures. However, Scherzer and colleagues demonstrated that only the stimulation efficiency increased with higher temperatures, whereas the mechanical properties of the sensory hairs were unaffected [20]. Therefore, an alternative explanation for the small differences in the sensory hairs' mechanical properties could be the distinct biological conditions of the traps during the deflection experiments. We performed all our experiments on intact, well-watered plants. In our hands, detachment or bisection of the traps quickly led to partial dehydration, and lower forces were necessary to deflect the sensory hair. Furthermore, data gained from experiments on partially dehydrated plants would be difficult to interpret because the sensory hairs would bend along the shaft upon deflection. In contrast, sensory hairs of fully hydrated plants bend mainly at the constriction whereas the shaft is hardly deformed (S2 Video). Nevertheless, the angular deflection and angular velocity values we measured are in the same order of magnitude not only with those obtained by Scherzer and colleagues using a force sensor but also with the values they measured when ants walked against the sensory hairs. This strongly indicates that our results reflect values that could occur in nature.

Using these parameters in a simple electromechanical model showed that single deflections of the sensory hair should be able to generate two APs, and thus lead to trap closure, a prediction that could be experimentally verified (Fig 4). In the observed cases, trap closure was not the result of the summation of subthreshold RPs from preceding nontriggering deflections, because the dwell time between the deflections was chosen long enough for the membrane potential to reset [23–25]. Moreover, we observed single trap closure without any prestimulation in several cases when the sensory hair was deflected with an angular velocity of 0.03 rad s$^{-1} \leq \omega \leq 4$ rad s$^{-1}$ (S3 Fig and S2 Video). Our results thus contradict the old dogma that two deflections are necessary to provoke trap closure but confirm that at least two APs must be elicited to trigger the snapping mechanism.

Taken together, our results show that an increasing strain on either side of the sensory hair's constriction is the determining factor for the generation of APs. We propose a mechanism in which the putative mechanosensitive channels in the plasmamembrane of the sensory cells at the constriction are open as long as they are stretched. Since this effect is counteracted by a membrane's natural tendency to relax (the global effect is seen in the sharp force decay in our sustained displacement experiment, Fig 1F), an increasing strain on the sensory hair is required to keep the channels open. As a consequence, the mechanosensitive channels will not open if the strain rate is too low (very slow deflection or sustained displacement). In a classical situation, when the deflection is fast and the angular displacement large enough (i.e., when an insect touches the sensory hair), the RP threshold is reached and a single AP is generated. In the case of a slower deflection, the RP threshold is reached during the deformation, eliciting an AP while the membrane is still under strain, thus reaching a second RP threshold during the same deflection. A second AP and thus trap closure are the consequence. Since the velocity range for single-touch snapping is far below that of the classical Venus flytrap prey, we can only speculate about the relevance of this observation in nature. However, one can imagine that it could be an advantage for catching slower prey animals, such as slugs, snails, or larvae, which may not touch a hair twice within a 30-s time span but for which anecdotal evidence suggests they can be prey.

## Materials and methods

### Venus flytraps

Our Venus flytrap population of about 100 plants was grown from seeds in 2011. The original seed batch was a donation from the Botanical Garden Zurich (https://www.bg.uzh.ch). Once a year, the plants were split and repotted into 9-cm clay pots, which were reused after they had been cleaned from moss. As a substrate, we used a mix of 90% white peat (Zürcher Blumenbörse, Wangen, Switzerland) and 10% granulated clay (SERAMIS Pflanz-Granulat, Westland Schweiz GmbH, Dielsdorf, Switzerland). The Venus flytraps were grown in a greenhouse at 60% relative humidity and a temperature regime of 18˚C–23˚C during the day and 16˚C–21˚C during the night. Plants were grown under normal daylight, morning and evening periods being extended by 400-W metal-halide lamps (PF400-S-h, Hugentobler Spezialleuchten AG, Weinfelden, Switzerland) to ensure a day length of 16 h. The lamps were also turned on when the daylight was insufficient. Rainwater was used to irrigate the plants whenever possible. For the experiments, plants were transported in an insulated moisture chamber from the greenhouse to the laboratory. All experiments were performed in vivo at temperatures between 19˚C and 21˚C. Plants were put back into the moisture chamber after every set of measurements.

### System configuration for force measurements

The system for force measurements consists of two subsystems, the first of which combines a force probe with a microrobotics system to quantify the sensory-hair deflection parameters. The second subsystem uses a load cell, metal levers, and hinges to measure the snap forces of the trap (Fig 2A and 2B). The second subsystem also serves to protect the force probe of the first subsystem from damage by preventing the trap from closing when the snapping mechanism was triggered. In addition, we attached the electrode for measuring APs to the metal levers, such that it was not necessary to reattach the electrode to each individual trap measured.

### Sensory-hair deflection

The deflection force sensor is a MEMS-based capacitive force sensor (FT-S1000-LAT; FemtoTools AG, Buchs, Switzerland) with a force range of ±1,000 μN and a standard deviation of 0.09 μN at 200 Hz, which measures the forces applied to a force probe ($50 \times 50$ μm) in $x$ direction. The force signal was recorded with a multifunction I/O device (NI USB-6003; National Instruments [NI], Austin, TX, United States of America). The deflection force sensor was mounted via a custom-made acrylic arm to an $xyz$ positioner (SLC-2475-S; SmarAct, Oldenburg, Germany) with a closed-loop resolution of 50 nm.

 The deflection force sensor was placed inside the trap with the force probe in front of a sensory hair guided by the optical feedback of two USB microscope cameras (Fig 2A and 2B). The view from USB microscope 1 (DigiMicro Profi; DNT, Dietzenbach, Germany) is used to position the probe laterally in the center of the hair and to extract the length of the sensory hair $H$ and the distance $h$ of the contact point of the sensor probe relative to the constriction (Fig 2C). For that purpose, a reference step of 500 μm was performed with the force probe in $z$ direction, and the pictures captured at start and end position were used to determine the distance-per-pixel value. USB microscope 2 (DigiMicro; DNT), oriented perpendicularly to the first one, was used to bring the force probe in close proximity of the sensory hair. Additionally, the force sensor could be automatically placed at a defined distance from the hair by finding contact using the force readout and then moving back to a defined position.

We defined a single deflection as a combination of back-and-forth angular displacement of the sensory hair caused by the advance and return of the force probe in $x$ direction (i.e., parallel to the midrib). This movement is parallel to the force-sensitive direction of the force probe and also ensures a perpendicular contact between force probe and sensory hair. The velocity of the sensor probe $v$ during deflection can be defined, and either a maximum linear displacement $d$ or a maximum force $F$ is set to define the maximum deflection (Fig 2D). The position control was used for the double deflections, where the angular displacement had to be increased incrementally between subsequent experiments. The force feedback mode was used in the single-deflection experiments to ensure maximum sensory-hair displacement without damaging the deflection force sensor.

The initial angular velocity $\omega$ is then given as

$$\omega = \frac{v}{h} \tag{1}$$

Furthermore, the angular displacement $\theta$ during the deflection is given by

$$\theta = \arctan\frac{d}{h} \tag{2}$$

The applied torque $\tau$ is given by

$$\tau = F \cdot h \tag{3}$$

Feedback control and data logging of forces and positions during the deflection were executed in LabVIEW.

## AP measurements

APs were recorded by connecting an electrode to one of the metal lever arms of the snap force sensor and inserting the reference electrode into the soil (Fig 2E). Insulating tape was used to cover most of the lever, only exposing the tip that is in contact with the trap leaf. A droplet of conductive gel (Compex Gel 250G, Digitec Galaxus AG) was applied between tip and leaf to guarantee a better contact area, stabilizing the readout, and increasing the signal. The voltage between the two electrodes was read out using an analog input module (NI USB-6003; NI, Austin, TX, USA).

## Snap force measurements

The snap force sensor consisted of two metallic lever arms, which transferred the force of the closing trap onto a load cell (31E Mid; Althen Sensors & Controls, Leidschendam, the Netherlands) with a load capacity of 50 N and a resolution of 75 mN (Fig 7A). Both arms were hinged at one end, thus opening and closing similar to a second order lever. The load cell was fixed onto one of the lever arms, while an adjustable screw was inserted into the other lever arm at the same distance $l_{cell}$ from the hinge. The gap between the two levers could then be adjusted by loosening or tightening the screw in order to measure flytraps of variable sizes and shapes. The snap force sensor was mounted on a labjack and placed on a manual $xy$ stage. This setup allowed the precise spatial positioning of the lever arms within the trap.

The relation between the force $F_{close}$ applied at the lever by a closing trap and the resulting force at the load cell $F_{cell}$ (Fig 7B) is given by

$$F_{cell} \cdot l_{cell} = F_{close} \cdot l_{trap} \tag{4}$$

where $l_{cell}$ and $l_{trap}$ are the distances from the hinge to the load cell and to the middle of the

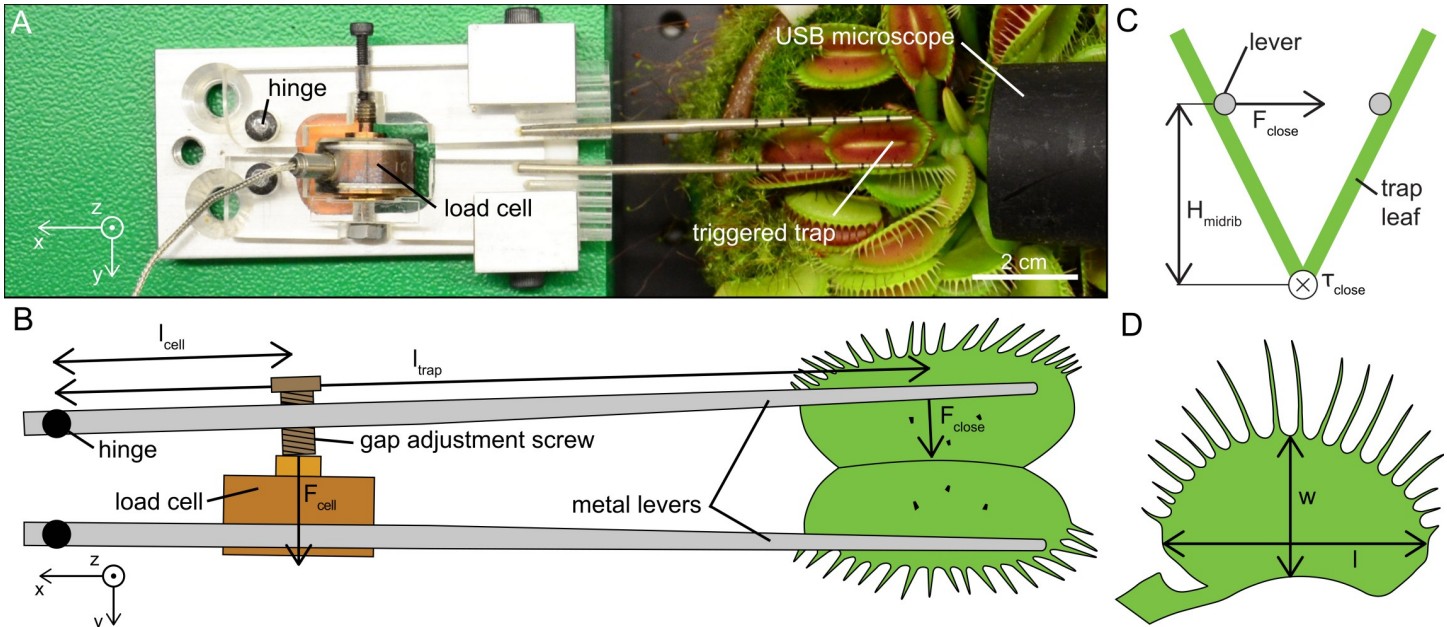

**Fig 7. Snap force sensor setup.** (A) Top view of the snap force sensor with a triggered trap. (B) Schematic top view of the snap force sensor showing how the force generated by the flytrap is transmitted onto the load cell. (C) Relation between measured force $F_{close}$, the height of the levers, and the closing torque $\tau_{close}$. (D) Schematic of the measured geometries, i.e., width $w$ and length $l$ of a trap.

trap leaf, respectively. The closing torque $\tau_{close}$ is given by

$$\tau_{close} = F_{close} \cdot H_{midrib} \tag{5}$$

where $H_{midrib}$ is the height between midrib and metal lever (Fig 7C). $H_{midrib}$ was determined using the ratios between width and length of the trap leaf (Fig 7D) and the top-down images of every individual experiment. The width and the length of 31 leaves were measured with a caliper to determine the ratio. The ratio $w/l$ of the leaves had a mean value of 0.59 and a standard deviation of 0.04. The mean value was used to calculate the torque for the remaining 16 experiments, in which we did not measure the traps' geometries.

The lever arms were carefully moved into the trap while maintaining an optimal gap between them to prevent contact with the sensory hairs. The gap between the levers was then increased until they contacted the surfaces of the lobes on both sides and underneath the marginal teeth. This configuration was maintained throughout the sensory-hair deflection experiments, in which it reduced perturbations by fixing and stabilizing the two lobes of the trap. By keeping the two lobes of the trap open even after closure had occurred, we could ensure that the deflection force sensor would not get damaged. The force signal was continuously recorded using a analog input module (NI USB-6009; NI, Austin, TX, USA).

Additionally, the time between mechanical stimulation and trap snapping (i.e., delay time) was determined by counting the frames recorded by the USB microscope between the second deflection of a hair and the first movement of the trap. The resolution of the snap force sensor with a reading frequency of 10 Hz would have been well suited for the delay-time analysis. However, since the snap force sensor and the deflection force sensor were not synchronized, we had to resort to the somewhat less accurate analysis of the videos.

## ECB model

To provide a comprehensive insight into the electromechanical events pertaining to the snapping of the Venus flytrap, an empirical ECB model is implemented, based on concepts of mechanotransduction and plant electrical memory. The model uses as input the angular velocity $\omega$ versus time $t$ profiles, which can comprise arbitrary waiting times between multiple stimuli. As output, it evaluates the number of APs that can be triggered via mechanical stimulation and further predicts the initiation of trap closure. Trap tissue is known to be capable of accumulating charges and firing APs when a threshold has been attained. Taking the findings of Volkov [32] and Jacobson [10], triggering of trap closure is a two-step procedure involving memory effects on different time scales—namely, the sensory memory (SM) and short-term memory (STM). By external injection of charges with a capacitor, the time scale of SM was found to range between 0.2 and 3 s, whereas STM of the trap was found to have an effect for several seconds to as long as a minute. As charge builds up in the SM, the RP increases until a charge threshold $Q_{RP}^{th}$ has been attained, and then an AP is fired. At the onset of an AP, the STM witnesses a charge buildup with simultaneous decay, and when the amount exceeds a threshold $Q_{STM}^{th}$, the trap was found to close [33]. By combining these findings with our own observations, we developed an empirical ECB model to provide a comprehensive picture of trap snapping.

The model comprises (1) simple mechanotransduction of the stimulus; (2) buildup and decay of the RP and the triggering of APs; and (3) accumulation and decay of charges in the STM with triggering of trap closure, when charges exceed a threshold (Fig 8). To incorporate the possibility of desensitization [20], an extension to sensitivity and charge transfer is made.

1. Mechanotransduction is implemented by relating the charge increment $\Delta Q_{RP}^{n+1}$ with $\omega$, by using a sensitivity multiplier $k$ that is reduced for increasing charges $Q_{RP}^{n}$ as in a feedback loop. Accordingly, the charge $\Delta Q_{RP}^{n+1}$ added at every step with angular speed $\omega(t)$ during the time increment $\Delta t$ reads

$$\Delta Q_{RP}^{n+1} = k \cdot \left( 1 - \frac{1}{\exp(-a(Q_{RP}^n - c))} \right) \cdot \omega(t) \cdot \Delta t. \tag{6}$$

The sigmoidal membership function provides a continuous desensitization for increasing $Q_{RP}^n$ with the parameters $a = 1.99$, $c = 3.48$, and $k$ is obtained as $k = 40.20$ μC rad$^{-1}$ to best describe the experimental results. When $\omega = 0$ rad s$^{-1}$ (i.e., when sensory hairs are simply held at a constant deflection), the RP does not increase. Furthermore, to avoid excessive charge buildup $\Delta Q_{RP}^{n+1}$ for very fast deflections, a limiting flux $\dot{Q}_{RP}^{lim}$ is required. This is regulated by taking the minimum of a sigmoidal flux limitation law

$$\Delta Q' = \frac{1}{1 + \exp(-5 \cdot \frac{\Delta Q_{RP}^{n+1}}{\Delta t \dot{Q}_{RP}^{lim}})} \tag{7}$$

and $\Delta Q_{RP}^{n+1}$ itself, where maximum charge flux is limited to $\dot{Q}_{RP}^{lim} = 93.2$ μC s$^{-1}$, identified experimentally through the limitation of snapping for high $\omega$. Note that advance and return phases for the hair deflection equally contribute to the charge buildup.

2. Resulting RP is continuously reduced because of charge diffusion via the exponential characteristic decay constant $\tau_{RP}$ [33] using the relation

$$Q_{RP}^{n+1} = Q_{RP}^n \cdot exp(-\Delta t/\tau_{RP}) + \Delta Q_{RP}^{n+1}. \tag{8}$$

The value of $\tau_{RP} = 0.64$ determines the limit for snapping at low $\omega$. If the threshold $Q_{RP}^{th} = 1$

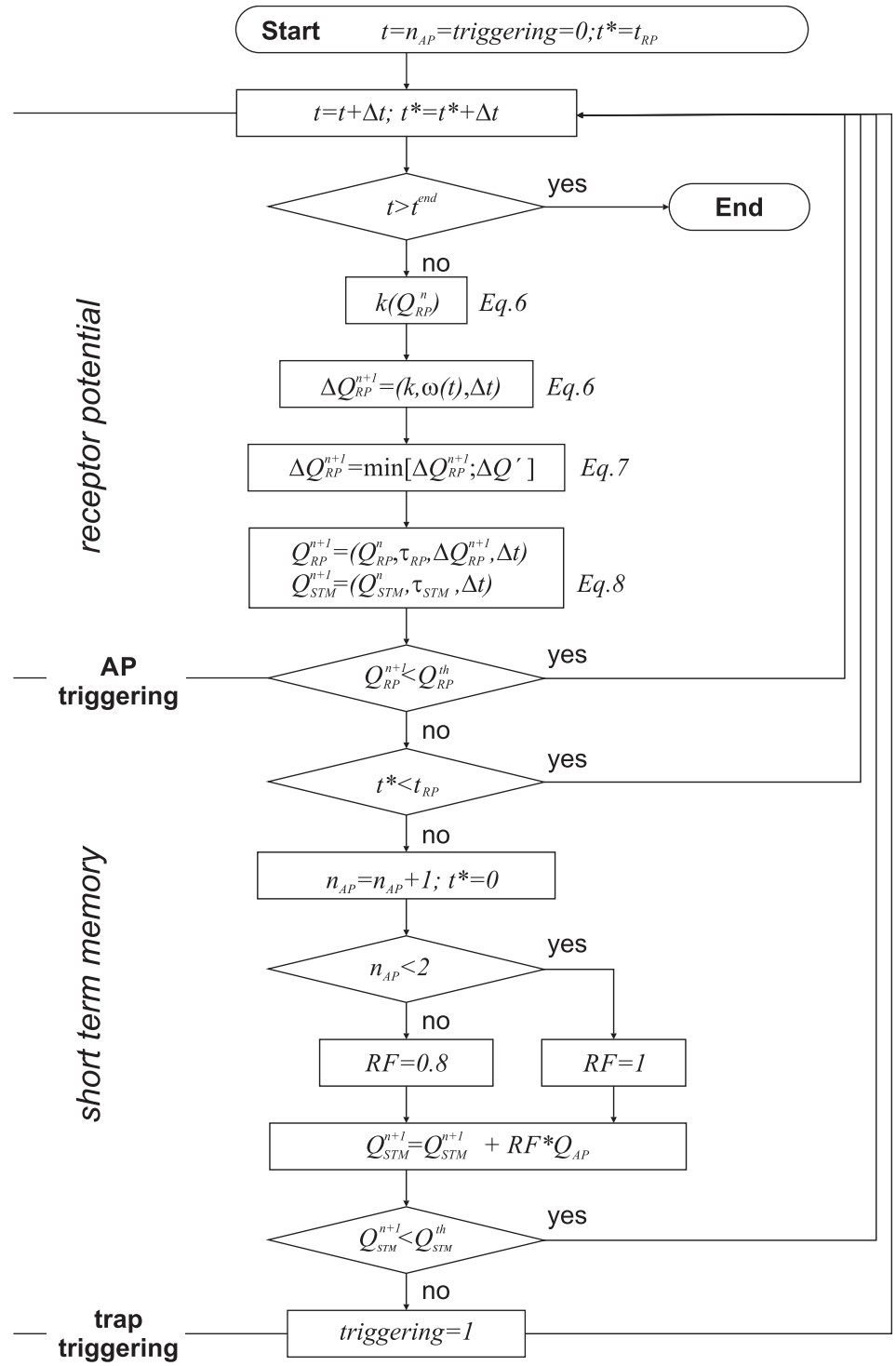

**Fig 8. Flow diagram of the model implementation.** $t$ denotes the total time of the experiment, that goes from $t = 0$ up to $t = t^{end}$, and $t^*$ is the relative time since the triggering of the last AP. Note that the reduction factor $RF$ is kept as $RF = 1$ for the simulations. AP, action potential.

for triggering an AP is reached, an AP is triggered. After an AP is triggered, RPs continue to increase if the stimulus continues, but an AP can only be fired after a constant refractory period $t_{RP}$ = 1.03 s [28].

3. After an AP has been triggered, a charge $Q_{AP}$ = 6 μC is transmitted to the STM. To meet measurements of Scherzer [20] for repeated triggering and desensitization, consecutive $Q_{AP}$s have the possibility to be reduced by a constant factor $RF$. For simplicity, we have configured $RF$ = 1. Note that in our experiments, it is unclear whether earlier deflections that did not result in trap triggering did or did not trigger APs. Charges in the STM again exponentially decrease (same as Eq 8) but with decay constant $\tau_{STM}$ = 39.71 s $> \tau_{RP}$. If the charge threshold for triggering $Q_{STM}^{th} = 1.67 \cdot Q_{AP}$ is reached, trap closure is initiated; however, the continuation of the stimulus for high RPs that did not decay below the threshold can trigger further APs after the refractory period. If the threshold is not reached, additional deflections are imposed to supplement $Q_{RP}$.

The model, therefore, only comprises the two decay constants for RP and STM ($\tau_{RP,STM}$), two respective thresholds ($Q_{RP,STM}^{th}$), the charge-dependent mechanotransduction rule, as well as refractory period ($t_{RP}$), maximum flux ($\dot{Q}_{RP}^{lim}$), and a reduction factor for consecutive APs. The last two parameters are not needed to represent the experimental data but to avoid unrealistic predictions for the full parameter space. Note that $Q_{STM}^{th}$, $Q_{RP}^{th}$, and $Q_{AP}$ are fixed a priori, whereas the other parameters ($\tau_{RP,STM}$, $t_{RP}$, $\dot{Q}_{RP}^{lim}$, $k$, $a$, $c$) are determined by a multidimensional optimization procedure using surrogate models [34]. The ideal parameter set is reached when (1) all experiments are correctly predicted, (2) the AP triggering sequence of Fig 4C is correctly calculated, (3) the maximum of double-deflection experiments trigger two APs, and (4) parameters are within reasonable physical and biological bounds.

## Bringing it all together—Predictions of the ECB model

Mechanical stimulation on the sensory hair is calculated for the angular velocity $\omega \in [0.01, 100]$ rad s$^{-1}$ and maximum angular deflection $\theta_{max} \in [0, 0.6]$ rad and comprises the advance and return phases, as well as resting periods of 1 s between repeated deflections. Opposite to experiments, the ECB allows predictions for the entire parameter space given by maximum angular deflection $\theta_{max}$ and deflection rate $\omega$. Depending on selected parameters, one finds regions where three, two, or even one deflection is sufficient to trigger trap closure, bordered by zones where no triggering can be achieved (see Fig 3). In all triggering zones, two APs are sufficient for trap closure; however, by further deflecting the hair, or even during resting periods, additional APs can be elicited. As a matter of fact, even the strongest possible desensitization that only depends on the charge of the RP cannot avoid multiple firing. Several procedures for additional desensitization in form of $Q_{STM}$ feedback or as function of $n_{AP}$ were implemented but, because of a lack of experimental insight, not further explored. Note that the single-deflection zone must be bounded by the multiple-deflection zones. When deflection rates are too small ($\omega < 0.04$ rad s$^{-1}$), RP buildup is not sufficient to make up for the charge decay. On the contrary, when $\omega$ is too large, the limited flux, as well as the refractory period $t_{RP}$, prevents the buildup of a critical charge amount. For multiple deflections, the resting period of 1 s between deflections exceeds $t_{RP}$ and, therefore, does not impact the charge buildup. On the contrary, the maximal flux limitation prevents triggering of APs with more than two deflections at high $\omega$. Independent of the number of deflections required, the model shows quite a broad zone of maximal sensitivity. However, one should consider that velocities below 1 rad s$^{-1}$ represent rather slow deflections.

In Fig 3, the parameters were selected as $\tau_{RP}$ = 0.64 s, $\tau_{STM}$ = 39.71 s, $Q_{RP}^{th}$ = 1 μC, $Q_{STM}^{th}$ = 10 μC, and $Q_{AP}$ = 6 μC, and $t_{RP}$ = 1.03 s, $\dot{Q}_{RP}^{lim}$ = 93.2 μC s$^{-1}$, and hair sensitivity of $k$ = 40.2 μC rad$^{-1}$ were selected within reasonable bounds to fit the experimental data for triggering by single and double deflections. Additionally, the optimization gave values for $a$ = 1.99, and $c$ = 3.48. The experimental points resemble the values for trap closure of angular deflection and angular velocity (see Fig 3). Using polynomial chaos expansion, we calculated the relative importance of the variance of each model input parameter for the variance of the model prediction accuracy. The resulting Sobol indices revealed a dominating influence of the sensitivity parameter $k$. Note that contrary to the model, in reality all parameters would be a function of temperature and exhibit a rather broad distribution.

The model explains the observed behavior in the entire parameter space and establishes the connection between the mechanical stimuli, APs, and subsequent trap closures, previously considered in separate ways [10, 20, 32]. Even though simplifying assumptions for relations and parameters were necessary, the available experimentally observed behavior can be explained consistently. The release of more than two APs per deflection is controlled by the charge dependence of the mechanotransduction rule and predicted with the parameters chosen for the model but will be addressed experimentally in the future.

## Supporting information

**S1 Fig. Double-deflection experiments.** (A) Torque versus angular velocity plot for the double-deflection experiments shown in Fig 1B. (B) Visualization of the force needed to sufficiently deflect a hair for double-deflection triggering. (C) Torque versus angular velocity plot for the single-deflection experiments shown in Fig 4B. For the underlying data, see sheets S1A Fig and S1C Fig in S1 Data.
(EPS)

**S2 Fig. Summary of all the mechanical stimuli that resulted in trap closure.** For the underlying data, see sheet S2 Fig in S1 Data.
(EPS)

**S3 Fig. Single deflections without any prestimulation of the sensory hairs can lead to trap closure.** (A-B) The traps triggered during bending (on the way in) and (C-D) triggered at the end of the return (on the way out). These experiments did not have preceding, nontriggering deflections and were therefore not included in Fig 4B. For the underlying data, see sheets S3A Fig to S3D Fig in S1 Data.
(EPS)

**S1 Video. Double deflection of a sensory hair leading to two APs and trap closure.** The sensor-probe position (motion in $x$ direction), the force at the deflection force sensor, and the measured APs are displayed next to the movie (Fig 1C is from this movie). AP, action potential.
(AVI)

**S2 Video. Single deflection of a sensory hair leading to trap closure.** At slow angular velocities, a single deflection is sufficient to trigger trap closure. The trap shown was not prestimulated, demonstrating that single-deflection closure is solely depending on the angular velocity at a sufficiently large deflection and is not the result of a summation effect due to previous subthreshold stimuli. This video also demonstrates that the sensory hair bends mainly at the constriction whereas the lever is hardly deformed. The trap does not move when the sensory hair

is deflected.
(AVI)

**S3 Video. Single deflection of a sensory hair leading to two APs shortly after one another during initial bending, resulting in trap closure.** The sensor-probe position (motion in *x* direction), the force at the deflection force sensor, and the measured APs are displayed next to the movie (Fig 4C is from this movie). AP, action potential.
(AVI)

**S1 Data. Raw data.** This file contains all the raw data underlying the plots presented in Figs 1, 3–5, and S1–S3 Figs.
(XLSX)

## Acknowledgments

We would like to thank Karl Huwiler and Christian Frey for taking care of the Venus flytraps and providing us with healthy plants.

## Author Contributions

**Conceptualization:** Hannes Vogler, Hans J. Herrmann, Bradley J. Nelson, Ueli Grossniklaus.

**Funding acquisition:** Hans J. Herrmann, Bradley J. Nelson, Ueli Grossniklaus.

**Investigation:** Jan T. Burri, Eashan Saikia, Nino F. Läubli, Hannes Vogler.

**Methodology:** Jan T. Burri, Eashan Saikia, Nino F. Läubli, Hannes Vogler, Falk K. Wittel, Markus Rüggeberg.

**Software:** Jan T. Burri, Eashan Saikia, Falk K. Wittel.

**Supervision:** Hannes Vogler, Falk K. Wittel, Markus Rüggeberg, Hans J. Herrmann, Ingo Burgert, Bradley J. Nelson, Ueli Grossniklaus.

**Visualization:** Jan T. Burri, Nino F. Läubli, Falk K. Wittel.

**Writing – original draft:** Jan T. Burri, Eashan Saikia, Nino F. Läubli, Hannes Vogler.

**Writing – review & editing:** Jan T. Burri, Hannes Vogler, Falk K. Wittel, Ueli Grossniklaus.

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
