## [Editor Report · Decision Letter 0]

22 Nov 2019

Dear Ueli, 

Thank you for submitting your manuscript entitled "The mechanical basis for snapping of the Venus flytrap, Darwin’s ‘most wonderful plant in the world’" for consideration as a Research Article by PLOS Biology.

Your manuscript has now been evaluated by the PLOS Biology editorial staff as well as by an academic editor with relevant expertise and I am writing to let you know that we would like to send your submission out for external peer review.

Please re-submit your manuscript within two working days, i.e. by Nov 26 2019 11:59PM.

Kind regards,

Ines

--

Ines Alvarez-Garcia, PhD

Senior Editor

PLOS Biology

Carlyle House, Carlyle Road

Cambridge, CB4 3DN

+44 1223–442810

---

## [Decision Letter · Decision Letter 1]

2 Jan 2020

Dear Ueli,

Thank you very much for submitting your manuscript "The mechanical basis for snapping of the Venus flytrap, Darwin's 'most wonderful plant in the world'" for consideration as a Research Article at PLOS Biology. Thank you also for your patience as we completed our editorial process, and please accept my apologies for the delay in providing you with our decision. Your manuscript has been evaluated by the PLOS Biology editors, an Academic Editor with relevant expertise, and by three independent reviewers, although we have to date only received reports from two of them; we will forward you the third one if it is sent to us belatedly.

As you will see, the reviewers find the manuscript interesting and important for the field, but they also raise several points that need to be clarified before we can consider the paper for publication. In light of the reviews (attached below), we are pleased to offer you the opportunity to address the comments from the reviewers in a revised version that we anticipate should not take you very long. We will then assess your revised manuscript and your response to the reviewers' comments and we may consult the reviewers again.

We expect to receive your revised manuscript within 1 month.

**IMPORTANT - SUBMITTING YOUR REVISION**

*NOTE: In your point by point response to to the reviewers, please provide the full context of each review. Do not selectively quote paragraphs or sentences to reply to. The entire set of reviewer comments should be present in full and each specific point should be responded to individually.

*Resubmission Checklist*

*Published Peer Review*

*PLOS Data Policy*

*Blot and Gel Data Policy*

Best wishes,

Ines

--

Ines Alvarez-Garcia, PhD

Senior Editor

PLOS Biology

Carlyle House, Carlyle Road

Cambridge, CB4 3DN

+44 1223–442810

Reviewers’ comments

Rev. 1:

This is a highly interesting study of how the Venus flytrap closure is triggered. Is main strength is the simultaneous measurement of the force applied to the sensory hair, of the electric potential, and of the force associated with trap closure. However, the study and the writing suffer from a number of weaknesses. The data could be better used and the presentation could be greatly improved.

MAJOR

1) The authors make the implicit assumptions that sensing occurs at the base of the hair, that the hair remain straight, and that angular velocities at the base can be computed using displacement and distance from base. What is the evidence that angles are more relevant? (For instance, experiments on the same hair at different distances from the base.) How straight are hairs during force application? (One of the movies gives the impression that the hair bends.)

2) There is some inconsistency about what is sensed. It is stated based on preliminary experiments that the flytrap is sensitive to angular velocity, whereas Fig 1E implies that a total angular displacement is sensed. The authors could make a better use of their data on closure triggered by single deflection to check whether closure is better predicted by cumulated angle, by cumulated absolute velocity, or by another combined quantity.

3) What are the variations with velocity/angle of the delay between APs and closure? Is this predicted by the model?

4) If mechanosensitive channels are involved, how do the authors explain that both positive and negative angular velocity contribute to closing? This should be a discussion point.

5) What is the precision of the different measurements?

6) What is the justification for the forms of the equations used in the model? (Types of functions chosen, etc.)

7) How were model parameters chosen? Is it eye-balling or was a systematic fitting procedure applied (the former is not acceptable). How sensitive are the results to parameter values?

MINOR

1) Acronyms are overused.

2) Captions are not always sufficiently detailed.

3) How comparable are the operating velocity and natural velocities associated with preys?

4) Figure 2B is confusing. Perhaps specify "Number of deflections to trigger closure".

Rev. 2:

The authors study the touch sensitivity of the carnivorous Venus flytrap in the context of animal prey recognition. This is a topic of great interest to the reader of PLOS Biology. The authors have developed an electromechanical model from which they suggest that, under certain circumstances, one touch is sufficient to generate two action potentials. This model challenges findings well documented by other labs that two APs are required to close the trap. Do the author's experiments support their hypothesis?

General comment:

A recent Scientific Reports paper by Poppinga et al. discussing the two trigger hair biomechanics and -electrics papers published

Scherzer, S. et al. Venus flytrap trigger hairs are micronewton mechano-sensors that can detect small insect prey. Nat Plants. 5, 670-675 (2019).

and the bioRxiv version

Burri, J. T. et al. The mechanical basis for snapping of the Venus flytrap, Darwin's 'most wonderful plant in the world'. bioRxiv. https://doi.org/10.1101/697797 (2019).

of the PLOS Biology paper under discussion side by side.

Why the authors did not discuss the findings of Scherzer et al. 2019 publicly available since some time?

Questions/Answers:

Authors: Our results confirm the model's predictions, suggesting that the Venus flytrap may be adapted to a wider range of prey movement than previously assumed.

Question: What is the evolutionary advantage of a system able to respond to a single touch that is equivalent to background noise?

Comment: The authors have not addressed the fact that Dionaea can catch animals of different size from large and small traps, with the latter growing trigger hairs that are more sensitive to force (Ref 18).

Authors: The mechanical basis for snapping of the Venus flytrap, Darwin's `most wonderful plant in the world'

Comment: Why is Darwin's statement in the title and first sentence of the introduction in a paper on biomechanics and electrics of the trigger hair that very likely is not understood by general readers of PLOS Biology?

Authors: .. starting with the observations made by Moses A. Curtis in the 1830s [2].

Comment: The observations that Moses A. Curtis made should be stated clearly.

Authors: Previous attempts to correlate the mechanical stimuli to the generation of APs … were not quantitative, or the experiments were done in vitro using dissected traps [18].

Comment: When reading Ref 18, this statement appears incorrect and misleading. In contrast to the authors' statement, experiments in Ref 18 were conducted with intact Dionaea plants.

Authors: To overcome these shortcomings, we used a microelectromechanical (MEMS)-based force sensor mounted on a microrobotics system to precisely control the velocity and amplitude of the deflection and to simultaneously measure the applied force in vivo

Comment: Why was it necessary to measure the snap force and repeat experiments performed by Volkov et al., 2012? What extra information could be gained compared to a simple video observation of the closing movement?

Most importantly, the experimental setup used to measure snap force has likely interfered with the recording of surface action potentials, because two metallic arms were in direct contact with the inner surface of both trap lobes (line 251). The tissue of the inner trap lobe is electrically excitable and action potentials can be generated just by mechanical stimulation of its surface. Thus, any small movements of the trap relative to the metallic levers (e.g. due to vibrations, particularly during the high accelerations involved in the piezo movements) can stimulate the trap (at a different location), leading to complex artefacts.

Authors: Reasoning that hair deflection induced by spiders, ants, and flies, i.e. the 'classical' prey of D. muscipula [19], would be rather quick, we operated the microrobotics system at full speed to simulate these stimuli, resulting in high initial angular velocities ranging from 10 to 20 rad s-1.

Question: To understand the authors' reasoning, readers need to know how fast typical Dionaea prey move and how far trigger hairs get bent upon collision with typical prey animals. The angular velocities used here are actually higher than the range observed for deflections of running ants in Ref 18 (14.5 to 448°/s, corresponding to 0.25-7.8 rad/s).

Authors: If the trap did not close, we waited for 2 min to make sure that the RP was completely reset.

Comment: Surface electrodes in general and those the authors used in particular do not monitor the receptor potential, not even the membrane potential (Volkov, 2018).

Moreover, the authors have not considered several important early publications, which contain detailed results relevant to the issue in question:

- Macfarlane, J. M. (1902). Contributions to the history of Dionaea muscipula Ellis. Contr. Bot. Lab. Univ. Pennsylvania I, 1, 7-44.

- Brown, W. H. and Sharp, L. W. (1910). The closing response in Dionaea. Bot. Gaz. 49, 290-302.

- Brown, W.H. (1916) The mechanism of movement and the duration of the effect of stimulation in the leaves of Dionaea. Am. J. Bot. 3, 68-90

The 3 independent studies mentioned above, conducted extensive series of repeated trigger hair stimulations, with varying pauses between them. The different authors agreed that at least two mechanical stimuli are necessary to produce closure of Dionaea leaves. It was found that trap closure can occur for consecutive trigger hair stimulations even when these were separated by pauses of 2 min or longer (although the number of stimuli required for trap closure increased as the time period between successive stimuli was expanded). This means that the 2-min recovery time used in this study was clearly insufficient for resetting the sensory system. Hence, what is interpreted as trap closure after one deflection might actually represent the result of the cumulation of consecutive subthreshold trigger hair stimulations that finally pass the threshold.

Authors: Then the procedure was repeated with increasing angular displacements until the snapping mechanism was triggered (Fig 1C and 1D), which was the case when a median displacement threshold of = 0.18 rad or a median torque threshold of = 0.8 μNm (n=21) was reached (Fig 1E and S1A Fig). Therefore, an animal has to apply twice a force F around 0.5mN close to the tip of the sensory hair, or up to 5mN close to the constriction (S1B Fig) to trigger closure.

Question: The same question as before: are these numbers relevant to the trigger hair-prey interactions observed in real life? Are these forces in a physiological range?

The sensitivity reported here is lower than that recorded in ref 18 (mean thresholds for hair displacement in large traps: 0.078 rad and torque: 159.5 nNm).

Could the result the authors have recorded, result from rather artificial experimental protocol conditions, in particular high angular velocities? (what were the actual angular velocities for these tests?). Maybe higher forces are found because of an unfixed trap. If stimulation moves the entire trap as well as the sensory hair, inaccurate values will result.

Authors: When the two consecutive deflections were well below the displacement threshold, we never observed an AP. For deflection amplitudes near the displacement threshold, a single AP was elicited after the second deflection. This indicated that both deflections contribute to the RP, and that the AP induction threshold was only reached with the second deflection.

Question: Without having measured the membrane potential of the mechanosensitive AP-generating TH cells, the authors are not allowed to come to conclusions about electrical phenomena in the actual trigger hair. Thus, statements remain pure speculations.

Authors: .. we deflected the sensory hairs far beyond the angular displacement threshold and kept that position for 30 s (Fig 1F).

Comment: This situation is very artificial. What should readers learn from the outcome of the experiment about the biology of trigger hair-prey interactions?

Authors: If sustained displacement had contributed to the RP, it should have remained above the threshold, in which case we would expect a series of APs.

Comment: This is pure speculation again, because the authors have used surface potential electrodes only rather than placing a sharp voltage recording microelectrode in the trigger hair bending zone (Benolken et al. 1970 and Hodick, Sievers. 1988) and thus have no clue about triggered membrane potential changes in the mechanosensor. Therefore the statement ".. the deflection rate plays a pivotal role to build up the RP while static deflections do not contribute" is too far-fetched and in stark contrast to previous work (Benolken et al., 1970; Volkov et al., 2008). Moreover, the ECB model would benefit from high-resolution membrane potential recordings able to resolve the charge increment ΔQ and the RP and thus the threshold for closure.

Authors: Based on these findings, we developed a simple model to explore the limits of angular displacement and velocity within which the traps would react. In our electromechanical charge buildup model, a mechanical deflection leads to a charge buildup of the RP ..

Comment: What is the value of a model on the behaviour of the RP without having measured it?

All the authors did is measure the trap surface potential that is no more nor less than a qualitative proxy of the AP generated in the TH.

Authors: If the accumulated charges surpass a certain threshold value, an AP is elicited.

Comment: This fact is neither in question nor new, but was already documented by papers from the Volkov lab before. By the way, induction of AP in response to charges accumulated that pass a certain threshold is not a trigger hair-specific feature but applies to, for example, the midrib as well (Volkov et al., 2008).

Authors: .. we observed trap closure induced by single deflections at lower angular velocities. To narrow down the range in which this occurs, we repeatedly deflected the same sensory hair with varying angular velocities until the trap closed.

Comment: This experimental scenario again was not linked to any kind of trigger hair-prey collision observed in vivo. What does the reader learn from the results obtained? How many action potentials were triggered by these single deflections? Only two, such as in experiments shown in Fig 3C/D? Were 2 APs always triggered before the trap closed? This information should be provided for all experiments.

Authors: Between two consecutive deflections, we waited 2 min for the trap to recover and any RP to dissipate.

Comment: (see comment above)

This is again pure speculation, because the authors have not measured the TH RP.

Authors: .. the closing torque around the midrib close with a median of 0.65mNm is a better quantity to characterize the trap's closing force (Fig 4C). The delay time, i.e., the time between the mechanical stimulus and the start of trap closure, was 0.6 0.3 s (mean s.d., n = 18), which is in the same range as the 0.4 s previously reported [25].

Comment: The time resolution of the force sensor is probably not too well suited to mechanical stimulus and the start of trap closure. Why not compare delay times with those obtained with non-invasive optical methods?

Authors: In a natural situation, when the deflection is fast and the angular displacement large enough, i.e. when an insect touches the sensory hair, the RP threshold is reached and a single AP is generated. In the case of a slower deflection, the RP threshold is reached during the deformation, eliciting an AP while the membrane is still under strain, thus reaching a second RP threshold during the same deflection. A second AP and trap closure are the consequence. Since the velocity range for single-touch snapping is far below that of the classical Venus flytrap prey, we can only speculate about the relevance of this observation in nature.

Comment: In the Discussion the authors themselves conclude that their experimental settings were non-natural. So why should the reader bother about a situation that is not natural?

Authors: ... either a maximum force F or a maximum linear displacement d is set to define the maximum deflection (Fig 6D).

Comment: For which experiments was force feedback control used? No data appear to be shown in the paper. What was the angular velocity in this case?

Authors: Video S2.

Comment: It is not clear how angular deflection was extracted from this experiment. First, it appears that the force transducer and the direction in which it moves are not perpendicular to the axis of the hair. In the beginning of the deflection, the sensor can be seen moving to the left (and probably downward), deflecting the trigger hair to the left, but at 11 sec, the trigger hair snaps off the sensor tip, resulting in a movement further to the left (and probably upward). Second, the deflection is obviously influenced by the closing movement of the trap but this is not reflected in the angular displacement trace in Fig.3C. Third, the point of force application seems to move slightly towards the hair tip during the experiment. Fourth, the video is not fully synchronised with the force trace (the trap movement at 3.8 sec can be seen in the video before the change in the force).

Authors: Figure 3A,B

Comment: It seems that amplitude and angular velocity were varied simultaneously in these experiments, making it difficult to study the effects of one particular factor.

Authors: Fig 1C - 2 consequent touches were carried out within one second, and the trap closure was examined.

Comment: I wonder about the high frequency of stimulation. Since the AP lasts over one second (see Fig 1 C 193 s) is it ok to stimulate faster? Ref 18 reported desensitization at frequencies above 0.25Hz (one touch every 4 s). What is the physiological stimulation frequency?

Authors: line 67: Volkov Ref 20

Comment: this is supposed to be #23.

- line 97: ´the higher boundary was`…

Comment: Where are the data shown? Is the setup able to precisely perform such quick and far movements?

- line 98: ´Additionally, we performed another set…`

Comment: Where can the reader find the data?

Conclusion:

Most of the findings the authors present are not new.

Confirmatory results are, however, not discussed in the context of previous findings by other labs.

The experiments performed to support a model scenario of one strike gaining 2 APs appear ill defined, and artefacts resulting from insufficient pauses between stimulations, and interference from the snap force measurement have not been considered.

The authors concluding about the RP of the trigger hair AP generating cells without measuring the membrane potential is a No-Go in bioelectrics.

---

## [Decision Letter · Decision Letter 2]

13 Mar 2020

Dear Ueli,

Thank you very much for submitting a revised version of your manuscript "The mechanical basis for snapping of the Venus flytrap, Darwin's 'most wonderful plant in the world'" for consideration as a Research Article at PLOS Biology. This revised version of your manuscript has been evaluated by the PLOS Biology editors, the Academic Editor and the two original reviewers.

As you will see, the reviewers seem mostly convinced by the experiments that you have added to the revision. Reviewer 1 would like you to include to the Discussion a couple of points you make in the rebuttal, whereas Reviewer 2 raises several issues that remain unclear. In addition, this reviewer recommends you to add an experiment to show single-touch triggering of plants with no stimulation history (point 7). After discussing the reviews with the Academic Editor, we have decided not to insist in this last experiment and make it optional for publication. Please address/discuss the other points raised.

We anticipate the revision should not take you very long. We will then assess your revised manuscript and your response to the reviewers' comments and we may consult the reviewers again.

We expect to receive your revised manuscript within 1 month.

**IMPORTANT - SUBMITTING YOUR REVISION**

*Resubmission Checklist*

*Published Peer Review*

*PLOS Data Policy*

*Blot and Gel Data Policy*

Sincerely,

Ines

--

Ines Alvarez-Garcia, PhD

Senior Editor

PLOS Biology

Carlyle House, Carlyle Road

Cambridge, CB4 3DN

+44 1223–442810

Reviewers’ comments

Rev. 1:

The authors have addressed most of my concerns. Additional points about the response:

"Using angles allows us to normalize for different contact heights and different sensory hair geometries. We have good evidence that the sensory hair bends mainly at the constriction, whereas the shaft is hardly deformed."

This needs to be stated explicitly in the manuscript.

"Using angles allows us to normalize for different contact heights and different sensory hair geometries. We have good evidence that the sensory hair bends mainly at the constriction, whereas the shaft is hardly deformed. "

Please include this as a brief discussion point.

Rev. 2:

The sensory biology of plants in general and carnivorous plants in detail will attract the broad readership of PlosBiology.

Authors rebuttal: Our mechanical data are largely in agreement with those recently published by Scherzer and colleagues, ..

Comment: The unexpected new finding of the authors is that a trigger hair of Dionaea has to be touched only once to close the snap trap.

The author's observation, however, not only "contradict the old dogma that two deflections are necessary to provoke trap closure" but also the very similar study that was published in Nature Plants recently. The reader of the PlosBiology paper thus expects the finding of "one strike is enough" to be sufficiently supported by experimental evidence. In response to the initial version, this referee asked - because of the lack of experimental data - for the authors to tone down assumptions about the MS channels and behaviour of the resting potential (RP). Furthermore, the authors were asked to increase the delay in stimulation to more than just 2 min and in an independent test series use traps that were not stimulated before. The authors, however, have ignored these reasonable suggestions completely. Since the authors' studies were not submitted to a journal specialising in biomechanics/biophysics but PlosBiology, the reader of a biology journal needs to see the new "one touch is enough" finding discussed in a biological and evolutionary context.

Detailed comments and questions:

Title:

1. Comment: To get informed about the new findings right away, the reader should learn already from the title that mechanical and electrical properties of the trigger hair allow the trap to close after one stimulation

Abstract:

Authors: Our results confirm the model's predictions, suggesting that the Venus flytrap may be adapted to a wider range of prey movement than previously assumed.

2. Comment: The readers have fund situations where the trap closes after one stimulation. Is 1 the "wider range" to 2? Does the model propose situations where more than 2 touches would be required to close the trap? In this context it should be mentioned that

in their PNAS paper Escalante-Pérez et al. (2011) showed Dionaea under water stress required more than 2 touch APs to close the trap.

Introduction:

Authors: Although there is a general agreement that sensory hair deflection opens mechanosensitive ion channels, such channels have not yet been identified [17, 18].

3. Comment: In Arabidopsis and genomes of other plants as well MSLs and OSCAs have been identified as mechanosensitive channels of different force sensitivity and ion (Cl- vs Ca2+) selectivity. Without knowing which MS channels of which properties are expressed in the trigger hair of Dionaea, how could the authors derive a proper model on touch induced APs?

Authors: Previous attempts to correlate the mechanical stimuli to the generation of APs suffered from the lack of appropriate instrumentation [10, 19] and thus were not quantitative, or the experiments were done in vitro using dissected traps [20]. To overcome these shortcomings, we used a microelectromechanical (MEMS)-based force sensor mounted on a microrobotics system to precisely control the velocity and amplitude of the deflection and to simultaneously measure the applied force in vivo (Fig 1A and 1B, and 5).

.. and

In addition, using a non-invasive method, we measured APs to test the deflection conditions under which they are generated.

4. Comment: this statement is wrong! Ref. 20 had used a "system to precisely control the velocity and amplitude of the deflection and to simultaneously measure the applied force in vivo". Cutting of one lobe from the trap on an otherwise intact plant the authors cannot classify "in vitro". The reader also should not be given the wrong impression that Jacobson and Benolken [19] having performed with excised (ex planta) trigger hairs represent in vivo (in planta) studies.

Results:

5. Receptor potential: The paper is so imprecisely written in many places that the general reader and even reviewer #1 believe that receptor potentials are measured after all.

When mentioning receptor potential in Results, the authors should clearly state that they in their study did not measure receptor potential features but assume it.

Without measuring RPs under their experimental conditions, the authors have to make clear what is fact and what is speculation when talking about RPs.

Authors: These results suggest that a fast deflection of the sensory hair increases the RP to a certain level, which depends on the amplitude of the angular deflection. RPs can add up and may elicit an AP after several deflections if they are below the deflection threshold.

6. Comment: How come the authors think that a voltage drop of certain amplitude suffices to initiate an AP in the trigger hair podium? In plants, electrical signals are associated with cytoplasmic Ca2+ signals. In the giant alga Chara system, mechanical stimulation elicits RPs as well as APs, after 50 years of research it is not yet clear how mechanistically the RP translates into an AP but clearly shows that cytoplasmic Ca2+ is a key player (Shepherd et al. 2008 and papers cited).

Authors: Although a summation effect was reported in these publications, at least five deflections spaced by 2 min were necessary to induce trap movements, which were always slow and/or partial [23-25]. After the dwell time, the procedure was repeated with increasing angular displacements until the snapping mechanism was triggered (Fig 1C and 1D).

7. Comment: Using traps without a pre-stimulation history the authors should show that the trap can close on one applying an above critical angular displacement. The result of this experiment will support or falsify the authors prediction and thus is a must.

At the IPBC congress a poster (provided to the PlosBiology editor) was presented showing that Dionaea plants were generated that express the genetically encoded cytoplasmic Ca2+ sensor. With a single touch the Ca2+ in the bending zone of trigger hair podium rises but stays blow the threshold required for trap closure. Upon the 2nd touch Ca2+ level passes the threshold and the trap closes. The authors have not observed that a single touch causes Ca2+ to rise to an extent that is able to close the trap.

Discussion:

Authors: We performed all of our experiments on intact plants. In our hands, detachment or bisection of the traps quickly led to partial dehydration and lower forces were necessary to deflect the sensory hair.

8. Comment: Such statement without having presented data is a no-go in science. Authors are asked to describe in detail the experimental procedure, documenting what the authors call "partial dehydration" and showing statistically significant results. From the information provided, interested readers must be able to repeat the experiment and confirm the findings published in Plos Biology.

When showing and discussing the so far "not shown data" the authors may explain to the reader how partial dehydration could increase the mechanical sensitivity of the trigger hair. Why? In the PNAS paper Escalante-Pérez et al. (2011) observed the opposite phenomenon. When Dionaea suffers from drought "partial dehydration" the capacity for trap closure is reduced. In Forterre et al. Nature (2005) even excised traps were well suited kinematic and mechanical measurements that provided the basis to model the biomechanics of trap closure. So far there was no study showing that intact traps on the plant and those with one lobe removed respond differentially to mechanical stimulation.

Authors: Since the velocity range for single-touch snapping is far below that of the classical Venus flytrap prey, we can only speculate about the relevance of this observation in nature. However, one can imagine that it could be an advantage for catching slower prey animals, such as slugs, snails or larvae, which may not touch a hair twice within a 30 s time span but for which anecdotal evidence suggests can be prey.

9. Comment: this statement cannot go without proper citation. How often if at all have slugs, snails or larvae been found in closed flytraps in their natural environments? YouTube videos showing all kinds of small animals could be offered as meal for Dionaea, is not of any biological relevance.

---

## [Editor Report · Decision Letter 3]

22 Apr 2020

Dear Ueli,

Thank you for submitting your revised Research Article entitled "A single touch can be enough: quantitative measurements of mechanical parameters shed light on the snapping mechanism of the Venus flytrap" for publication in PLOS Biology. I have now obtained advice from the original academic editor and discussed the revision with the team of editors.

We're delighted to let you know that we're now editorially satisfied with your manuscript. However before we can formally accept your paper and consider it "in press", we also need to ensure that your article conforms to our guidelines. A member of our team will be in touch shortly with a set of requests. As we can't proceed until these requirements are met, your swift response will help prevent delays to publication. Please also make sure to address the data and other policy-related requests noted at the end of this email.

*Copyediting*

*Published Peer Review History*

*Early Version*

*Submitting Your Revision*

Best wishes,

Ines

--

Ines Alvarez-Garcia, PhD

Senior Editor

PLOS Biology

Carlyle House, Carlyle Road

Cambridge, CB4 3DN

+44 1223–442810

DATA POLICY:

Many thanks for sending us the data underlying all the graphs. Please also ensure that figure legends in your manuscript include information on WHERE THE UNDERLYING DATA CAN BE FOUND (in both main and supplementary figures).

---

## [Editor Report · Decision Letter 4]

26 May 2020

Dear Dr Grossniklaus,

On behalf of my colleagues and the Academic Editor, Mark Estelle, I am pleased to inform you that we will be delighted to publish your Research Article in PLOS Biology. 

Early Version

PRESS 

Kind regards,

Alice Musson

Publishing Editor, 

PLOS Biology

on behalf of

Ines Alvarez-Garcia,

Senior Editor

PLOS Biology